

# Isolation and identification of endophytic fungi from *Conyza blinii* that exhibit antioxidant and antibacterial activities

Yujie Jia[1,*], Guodong Zhang[2,*], Qiqi Xie[1], Jiwen Tao[1], Tongliang Bu[1], Xinyu Zhang[1], Yirong Xiao[3], Zhao Chen[4], Qingfeng Li[1] and Zizhong Tang[1]

[1] College of Life Sciences, Sichuan Agricultural University, Ya'an, China
[2] Shanghai Minhang District Agricultural Products Quality and Safety Center, Shanghai, China
[3] Sichuan Agricultural University Hospital, Sichuan Agricultural University, Ya'an, China
[4] Ya'an People's Hospital, Ya'an People's Hospital, Ya'an, China
* These authors contributed equally to this work.

## ABSTRACT

**Background:** As a medicinal plant, *Conyza blinii* is known to contain a wealth of bioactive constituents, including flavonoids, terpenes, and triterpenoid saponins, which contribute to its anti-inflammatory and anticancer properties. Endophytic fungi, which symbiotically inhabit plant tissues, are recognized for their ability to synthesize bioactive metabolites analogous to those of their hosts. However, the potential of *C. blinii*-associated endophytes remains underexplored. This study aims to isolate and characterize phenols-producing endophytic fungi from *C. blinii*, evaluate their biological activities, and analyze their chemical components to provide new insights for drug development.

**Methods:** During the study, 20 endophytic fungi were isolated from *C. blinii*. The Folin-Ciocalteu method was used to screen for strains capable of producing phenolic compounds. To assess their bioactivity, ethyl acetate extracts of different concentrations were tested for antibacterial and antioxidant activities. Antibacterial activity was evaluated using minimum inhibitory concentration (MIC) determinations, while antioxidant activity was assessed through 2,2-Diphenyl-1-picrylhydrazyl (DPPH) radical, 2,2′-Azinobis-(3-ethylbenzthiazoline-6-sulfonic acid) (ABTS) radical, hydroxyl radical, and superoxide anion radical scavenging assays. Additionally, liquid chromatography-mass spectrometry analysis was conducted to quantify the active components in the extracts.

**Results:** Among the isolated 20 endophytic fungi, four strains successfully produced phenolic compounds, with the highest total phenolic content of 77.17 ± 1.93 mg milligrams of gallic acid equivalents per gram of extract (GAE/g). All ethyl acetate extracts from the endophytic fungi exhibited good antibacterial and antioxidant properties. Notably, *Fusarium circinatum* demonstrated exceptional antioxidant activity, with scavenging rates for DPPH and ABTS radicals reaching 94.28 ± 0.042% and 96.60 ± 0.017%, respectively. The ethyl acetate extract of *F. foetens* showed remarkable antibacterial effects against *Escherichia coli* and *Staphylococcus aureus*, with MIC values as low as 0.5 mg/mL. Furthermore, liquid chromatography-mass spectrometry (LC-MS) analysis revealed that *F. foetens* could produce various

Corresponding author
Zizhong Tang, 67031988@qq.com

high-value phenolic compounds, including tyrosol (626.1884 ng/mL) and homovanillic acid (369.15486 ng/mL), which hold potential pharmaceutical value.
**Discussion:** This study isolated 20 endophytic fungi from *C. blinii*, discovering that four strains, produced phenolic compounds with strong antioxidant and antimicrobial properties. Among them, *F. circinatum* exhibited the highest antioxidant activity. Additionally, the fungi produced bioactive metabolites with potential applications in health care, medicine, and agriculture. These findings highlight the potential of *C. blinii* endophytes for sustainable bioactive compound production.

## INTRODUCTION

*Conyza blinii*, an medicinal herb in southwestern China, has demonstrated significant anti-inflammatory properties through pharmacological studies (*Liu et al., 2017*; *Ma et al., 2017a*). Besides, it has a certain inhibitory effect on cancer cells (*Ma et al., 2017b*; *Peng et al., 2019*). It's main secondary metabolites that exert pharmacological effects mainly include flavonoids, terpenes, saponins and alkaloids, and the main active components are triterpenoid saponins (*Qiao et al., 2010*). However, escalating environmental pressures and anthropogenic disturbances have affected its natural populations and challenged the production of bioactive substances.

Endophytic fungi are harmless microorganisms that symbiotically inhabit plant tissues (*Banerjee, 2011*). In addition, endophytic fungi can synthesize secondary bioactive metabolites functionally equivalent to those of their host plants (*Bamisile et al., 2018*). These microbial-derived secondary metabolites mainly include alkaloids, aliphatic compounds, terpenes, steroids, flavonoids and phenols (*Aly, Debbab & Proksch, 2011*). These compounds demonstrate multifaceted bioactivities including antioxidant, antimicrobial, and antitumor effects (*Carrión et al., 2019*). Of particular significance is their superior fermentation efficiency relative to plant-based metabolite production systems (*Wu et al., 2018*). Hence, they represent a vast resource for the development of bioactive drugs, offering great potential for exploitation and application.

Free radical homeostasis is maintained through endogenous antioxidant defenses under physiological conditions (*Khojah et al., 2016*). Disruption of this equilibrium promotes oxidative damage to DNA, accelerating the aging process and contributing to a range of diseases (*Masisi, Beta & Moghadasian, 2016*). Emerging evidence indicates that endophytic fungal metabolites have a good scavenging effect on a variety of free radicals (*Ibrahim et al., 2021*; *Hoque et al., 2023*; *Hashem et al., 2023*). Therefore, endophytic fungi can serve as sustainable alternatives for natural antioxidant production.

In addition, the presence of certain pathogenic microorganisms can also threaten human health and the survival of other organisms (*Turner et al., 2018*). Existing studies have shown that certain metabolites produced by endophytic fungi show effective

antimicrobial bioactivity. For example, four endophytic fungal extracts from lotus inhibited *Staphylococcus aureus*, *Streptococcus mutans*, *Staphylococcus epidermidis*, *Pseudomonas aeruginosa* and *Propionibacterium acnes* (*Techaoei et al., 2021*). In addition, *Chaetomium globosum* exopolysaccharides from *Gynostemma pentaphyllum* demonstrated antibacterial activity on *S. aureus* and *E. coli*. (*Wang et al., 2023a*). These findings underscore the untapped potential of endophytic fungi in antimicrobial development.

It is important to highlight their significance, considering that medicinal plants and their associated endophytes comprise over 80% of the natural remedies available in the market (*Singh & Dubey, 2015*). More and more researchers are obtaining bioactive compounds with therapeutic activity by isolating and using endophytic fungi from medicinal plants. At present, research on the bioactive compounds produced by *C. blinii* endophytic fungi is limited. Therefore, the aim of this study is to isolate and identify endophytic fungi from *C. blinii* that produce substances with antioxidant and antibacterial activities, and to analyze the active compounds with potential application value using liquid chromatography-mass spectrometry (LC-MS). The bioactive compounds identified from *C. blinii* endophytic fungi demonstrate dual functionality—potent free radical scavenging capacity and antibacterial activity—which positions them as promising candidates for developing natural preservatives in the field of antioxidant foods or antimicrobial drugs.

# MATERIALS AND METHODS

## Experimental materials

In November 2022, *C. blinii* were randomly collected from the Sichuan Agricultural University farm in Ya'an, Sichuan, China (29.98°N, 102.99°E). Five mature and disease-free plants were selected and separated into root, stem, and leaf samples for further use. After being placed in the sampling bag, each sample was submitted straight away to the lab for analysis within 12 h.

## Isolation of endophytic fungi

The methods for isolating endophytic fungi from plants are mainly based on *Zhao, Xu & Jiang (2012)*, as outlined below. Samples from different plant parts were washed with water. The obtained samples were cut into small cubes, each measuring approximately 5 mm, and washed meticulously with sterile water five times. The samples were immersed in a 75% ethanol solution for 1 min, then treated with 5% sodium hypochlorite for 8 min, and lastly exposed to 70% ethanol for 0.5 min. The samples were washed three times with sterile water. Furthermore, the samples were gently wiped to remove excess liquid. The processed samples were placed on Potato Dextrose Agar (PDA) supplemented with 50 µg/L kanamycin sulfate and ampicillin (hereafter the same). After that, the PDA medium containing the samples was incubated at 28 °C until significant colonies occurred. Finally, different colonies were isolated and added to a new PDA medium until a single colony appeared on the PDA medium.

## Screening of phenols-producing endophytic fungi

The methods for screening of phenols-producing endophytic fungi are mainly based on *Tang et al. (2021)*. Endophytic fungi was transferred from PDA into 50 mL of potato dextrose broth (PDB) and cultured at 28 °C with 200 rpm shaking for 7 days. Following cultivation, the fungal mycelium was collected, treated with sterile water, and the spore concentration was adjusted to approximately $1.0 \times 10^5$ spores/mL. Five mL prepared spore suspension was added to 200 mL of PDB and cultured under 28 °C, 200 rpm shaking for 7 days. The mixture was squeezed and filtered through cheesecloth to collect the fermentation broth. Endophytic fungi that produce phenols are initially recognized using a color reaction. The fermentation broth was combined with a chromogenic reagent (0.1% $FeCl_3$:0.1% $K_3[Fe(CN)_6]$ = 1:1) in a transparent tube. A blue hue change indicates the presence of phenols in the fermentation broth.

## Identification and phylogenetic analysis of phenols-producing endophytic fungi

Fungal gDNA (genomic DNA) was extracted with Rapid Fungi Genomic DNA Isolation Kit (Sangon Biotech (Shanghai)). General primers listed in Table S1 were employed to amplify the internal transcribed spacer (ITS) region of the fungal genome *via* polymerase chain reaction (PCR). The components of the PCR reaction system are detailed in Table S2, while the specific PCR reaction conditions are outlined in Table S3. The PCR results underwent direct sequencing by the TSINGKE Biological Technology Corporation in Beijing. After obtaining the sequencing results, the ITS region sequences were compared with known species sequences in GenBank. Phylogenetic trees were generated in MEGA X using the neighbor-joining technique to elucidate the relationships among endophytic fungal species.

## Preparation of fermentation products from endophytic fungi

The phenols-producing endophytic fungi were subjected to scale-up cultivation in PDB. Following filtration, 1,000 mL of filtrate was obtained from each endophytic fungus for extraction. The filtrate was divided into four separate 250 mL aliquots. Each aliquot was extracted twice with one of the following solvents: n-butanol, ethyl acetate, chloroform, or petroleum ether (two extractions per solvent, each performed on an independent 250 mL aliquot). The organic extracts were subsequently concentrated by rotary evaporation until no significant volume change was observed. Finally, the concentrated extracts were freeze-dried. The lyophilized crude materials were collected and weighed. The dry mass was dissolved in dimethyl sulfoxide (DMSO) and the concentration was adjusted to 10 mg/mL to create a stock solution for assessing the biological activity.

## Quantification of total phenolic content

The Folin-Ciocalteu (FC) assay was used to determine the fungal extracts' total phenolic content (TPC), using gallic acid as the standard reference compound (*Minussi et al., 2003*). Specific operations are as follows: the stock solution (0.5 mL), FC reagent (0.5 mL) and ddH$_2$O (0.5 mL) were blended and reacted for 1 min. Afterwards, 1.5 mL 20% Na$_2$CO$_3$

solution was added. Subsequently, the combination was diluted to a final volume of 10 mL with ddH$_2$O. After 10 min of heating in a water bath at 70 °C. The Multiskan Sky (Thermo Fisher Scientific, Waltham, MA, USA) was used to measure the absorbance at 760 nm (A$_{760}$). Taking gallic acid as the standard solution, the regression equation was obtained as follows: y = 0.0299x − 0.0069, where R$^2$ = 0.991.

## Antioxidant activity

The stock solution was prepared at six different concentrations with EtOH (0.2, 0.4, 0.6, 0.8, 1.0, and 3 mg/mL) to be used in different antioxidant activity tests: 2,2-Diphenyl-1-picrylhydrazyl (DPPH) radical scavenging, 2,2′-Azinobis-(3-ethylbenzthiazoline-6-sulfonic acid) (ABTS) radical scavenging, ABTS radical scavenging, hydroxyl radicals cavenging, and superoxide anion radical scavengin. All tests were conducted with vitamin c (Vc) as the positive control.

### DPPH radical scavenging activity

This experiment followed the procedure outlined by *Nuerxiati et al. (2019)*: 100 μL 0.2 mmol/L DPPH and 100 μL different concentrations of the extract were added as the experimental group. In the dark, the reaction was conducted for 30 min at room temperature. The absorption was measured at a wavelength of 517 nm (A$_{517}$). The scavenging rate was determined using the following formula:

$$\text{Scavenging rate (\%)} = [1 - (A_{517} - A_0)/A_{max}] \times 100\%$$

where A$_{517}$ represents the absorbance of experimental group solution; A$_0$ represents the absorbance of different sample solution in EtOH; A$_{max}$ represents the absorbance of DPPH solution in EtOH.

### ABTS radical scavenging activity assay

This experiment followed the procedure outlined by *Zhao et al. (2005)*: The ABTS storage solution was prepared by mixing 7 mmol/L ABTS (0.5 mL) with 140 mmol/L K$_2$S$_2$O$_8$ (88 μL). The ABTS storage solution was diluted with ddH$_2$O and adjusted the absorbance to 0.70 at 734 nm to prepare the working solution. In each cavity of a 96-well plate, 50 μL ABTS working solution were combined with 150 μL sample solution as the experimental group over 6 min. The absorbance was measured at a wavelength of 734 nm (A$_{734}$). The scavenging rate was determined using the following formula:

$$\text{Scavenging rate (\%)} = [1 - (A_{734} - A_0)/A_{max}] \times 100\%$$

where A$_{734}$ represents the absorbance of experimental group solution; A$_0$ represents the absorbance of different sample solution mixed with EtOH; A$_{max}$ represents the absorbance of ABTS working solution combined with EtOH.

### Hydroxyl radicals scavenging activity assay

This experiment followed the method outlined by *Smirnoff & Cumbes (1989)*: The reaction solution consisted of 50 μL 6 mmol/L salicylic acid-ethanol, 50 μL 6 mmol/L FeSO$_4$, and 50 μL 0.1% H$_2$O$_2$. Different concentrations of sample solutions (50 μL) were added to the mixture as the experimental group and allowed to react at 37 °C for 30 min. The

absorbance was obtained at a wavelength of 510 nm ($A_{510}$). The scavenging rate was determined using the following formula:

$$\text{Scavenging rate (\%)} = [1 - (A_{510} - A_0)/A_{max}] \times 100\%$$

where $A_{510}$ represents the absorbance of experimental group solution; $A_0$ represents the absorbance of the solution with ddH$_2$O instead of H$_2$O$_2$; $A_{max}$ represents the absorbance of EtOH as the solvent instead of the sample solution.

### Superoxide anion radical scavenging activity assay

This experiment followed the method outlined by *Wang et al. (2009)*: 0.5 mL test samples solution of varying concentrations was combined with 1.5 mL 0.05 mol/L Tris-HCl buffer (pH 8.2) and reacted at 25 °C for 30 min. Afterwards, 25 mmol/L pyrogallol (200 μL) was added and reacted 4 min at 25 °C. To terminate the reaction, 0.25 mL of 8 mol/L HCl was added. The substance's absorbance was detected at 325 nm ($A_{325}$). The scavenging rate was determined using the following formula:

$$\text{Scavenging rate (\%)} = [1 - (A_{325} - A_0)/A_{max}] \times 100\%$$

where $A_{325}$ represents the absorbance of experimental group solution; and $A_0$ represents the absorbance of a solution containing ddH$_2$O instead of pyrogallol; $A_{max}$ represents the absorbance of EtOH as the solvent instead of the sample solution.

## Antibacterial activity

### Minimum inhibitory concentration determination

The determination of the minimum inhibitory concentration (MIC) was referenced from *Molla et al. (2016)*. Four typical bacteria (*Escherichia coli*: ATCC25922, *Staphylococcus aureus*: ATCC6538, *Pseudomonas aeruginosa*: ATCC9027, *Bacillus subtilis*: ATCC6633) were selected to assess the MIC. Among them, *S. aureus* and *B. subtilis* are Gram-positive bacteria, while the other two strains are Gram-negative bacteria. The four types of bacteria were inoculated into sterile Luria-Bertani (LB) broth and then incubated at 37 °C for 24 h. The fungal extracts were diluted to different concentrations (0.2–4 mg/mL) in sterile LB broth. A bacterium sample of 10 μL was introduced into Eppendorf (EP) tubes containing samples of varying concentrations. After incubation at 37 °C for 24 h, color changes were observed using 10 μL 0.1 mg/mL methylthiazolyldiphenyl-tetrazolium bromide (MTT). A deeper color in the solution indicates a higher number of viable cells, which was used to determine the MIC.

### Minimum bactericidal concentration determination

The minimum bactericidal concentration (MBC) was assessed utilizing the method published by *Rocha et al. (2020)*: 10 μL bacterial culture, that was incubated at 37 °C for 24 h, was mixed with an equal volume of fungal extract at different concentrations. Then, the mixture was spread onto LB solid agar and incubated at 37 °C for 24 h. Any result showing fewer than five colonies on the plate was deemed to have a bactericidal effect.

### Analyzing compounds of endophytic fungal extracts using LC-MS

Analyzed endophytic fungi extracts using Ultra Performance Liquid Chromatography-High Resolution Mass Spectrometry (UPLC-HRMS) (Waters, UPLC; Thermo, Q Exactive). The specific chromatographic and mass spectrometry acquisition conditions are detailed in Tables S5, S6, S7. The MS data were analyzed using Compound Discoverer 2.0 in conjunction with the mzCloud, Metlin, and Human Metabolome Database (HMDB) databases.

### Data analysis

All experimental results were performed with three biological replicates. Using a significance level of $p < 0.05$, analysis of variance (ANOVA) was conducted using SPSS 24.0 software, followed by letter marking using the Waller-Duncan test. The significance level for statistical differences was set at $p < 0.05$ to determine the statistical differences among the various groups.

## RESULTS

### Preliminary separation of endophytic fungi from *C. blinii*

Twenty distinct endophytic fungi have been separated from leaves, stems, and roots and were numbered. Eight isolates were obtained from roots, seven from stems, and five from leaves. The endophytic fungi extracted from different tissues of *C. blinii* are listed in Table S4. The endophytic fungal colony cultured by PDA medium is shown in Figs. S1.

### Screening of phenols-producing endophytic fungi

The isolated endophytic fungi were subjected to a chromogenic reaction for the screening of phenols-producing fungi. After the reaction of $FeCl_3K_3[Fe(CN)_6]$ solution with the supernatant of endophytic fungi fermentation broth, as shown in Fig. 1. All fluids could react with chromogenic agents, demonstrating the strains' ability to generate phenols. Four strains (CBF5, CBF9, CBF10, CBF18) with rapid change and darker reaction color were chosen for further chemical and pharmacological analyses.

### Molecular identification of phenols-producing fungi

The results of PCR amplification products for each strain are shown in Fig. S2. Further, molecular determination of endophytic fungi was performed by rDNA sequence analysis based on their ITS sequence. Figure 2 displays the evolutionary tree of the endophytic fungus that produce phenols. The four phenols-producing endophytic fungi were classified into *Fusarium*. Finally, CBF5 was preliminarily identified as *Fusarium pseudoanthophilum*. CBF9 was preliminarily identified as *Fusarium foetens*. CBF10 was preliminarily identified as *Fusarium circinatum*. CBF18 was preliminarily identified as *Fusarium panlongense*.

### Determination of total phenolic content

Four different solvents were used to extract the phenols produced by four endophytic fungi from *C. blinii*, and TPC of different extraction were measured (Table 1). The TPC of these
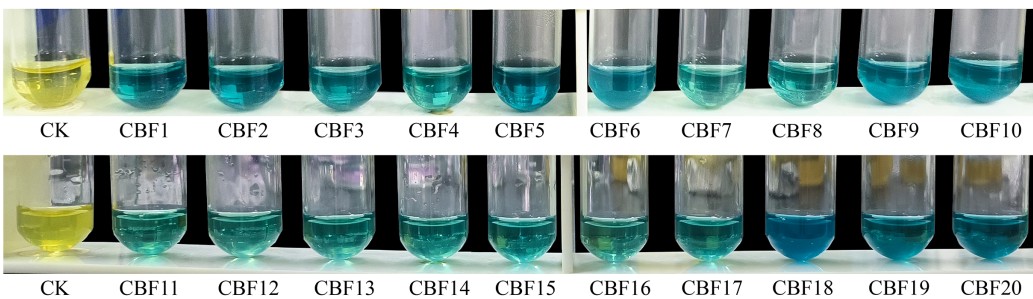

**Figure 1 Screening of phenols-producing endophytic fungi.**

extracts ranged from 3.75 ± 0.19 to 77.17 ± 1.93 mg GAE/g (milligrams of gallic acid equivalents per gram of extract). Moreover, The TPC of the ethyl acetate extracts from the four fungal species ranged from 60.45 to 77.17 mg GAE/g, all of which were significantly higher than those of the other three extracts. In summary, ethyl acetate allowed the extraction of more phenols.

## Antioxidant activity

To investigate the antioxidant activity of extracts, four different antioxidant indices (DPPH radical, ABTS radical, superoxide anion radical and hydroxyl radical) were measured to assess the antioxidant properties of four different fractions extracted from *F. pseudoanthophilum*, *F. foetens*, *F. circinatum*, and *F. panlongense*. Figure 3 displays the antioxidant activity of several extracts.

Figure 3 and Table 2 show the antioxidant activity values. As seen in Fig. 3, all extracts exhibited radical scavenging activity in a concentration-dependent pattern. As shown in Table 2, the scavenging activity of all extracts on four kinds of free radicals seemed to be related to the TPC. For the DPPH racial, the ethyl acetate extracts from various fungi exhibited good efficacy, among which the extract from *F. circinatum* showed the highest scavenging activity ($IC_{50\ DPPH}$ = 0.54 ± 0.007 mg/mL). It was followed by *F. foetens* ($IC_{50\ DPPH}$ = 0.56 ± 0.009 mg/mL) and *F. panlongense* ($IC_{50\ DPPH}$ = 0.55 ± 0.01 mg/mL). For the hydroxyl radical, the ethyl acetate extracts from various fungi exhibited good efficacy, among which the extract from *F. foetens* had the highest scavenging activity ($IC_{50\cdot OH}$ = 0.593 ± 0.020 mg/mL). It was followed by *F. pseudoanthophilum* ($IC_{50\cdot OH}$ = 0.854 ± 0.088 mg/mL) and *F. panlongense* ($IC_{50\cdot OH}$ = 0.840 ± 0.076 mg/mL). For the superoxide anion radical, the ethyl acetate extracts from various fungi exhibited good efficacy, among which the extract from *F. foetens* had the highest scavenging activity ($IC_{50}.o_2^-$ = 0.266 ± 0.013 mg/mL), followed by *F. circinatum* ($IC_{50}.o_2^-$ = 0.748 ± 0.035 mg/mL). For the ABTS radical, the n-butanol extract of *F. pseudoanthophilum* had the highest scavenging activity ($IC_{50\ ABTS+.}$ = 0.009 ± 0.005 mg/mL). It was followed by the ethyl acetate extract of *F. pseudoanthophilum* ($IC_{50\ ABTS+.}$ = 0.012 ± 0.004 mg/mL) and the n-butanol extract of *F. circinatum* ($IC_{50\ ABTS+.}$ = 0.026 ± 0.002 mg/mL). In summary, all extracts had higher ABTS radical scavenging. When the concentration of ethyl acetate

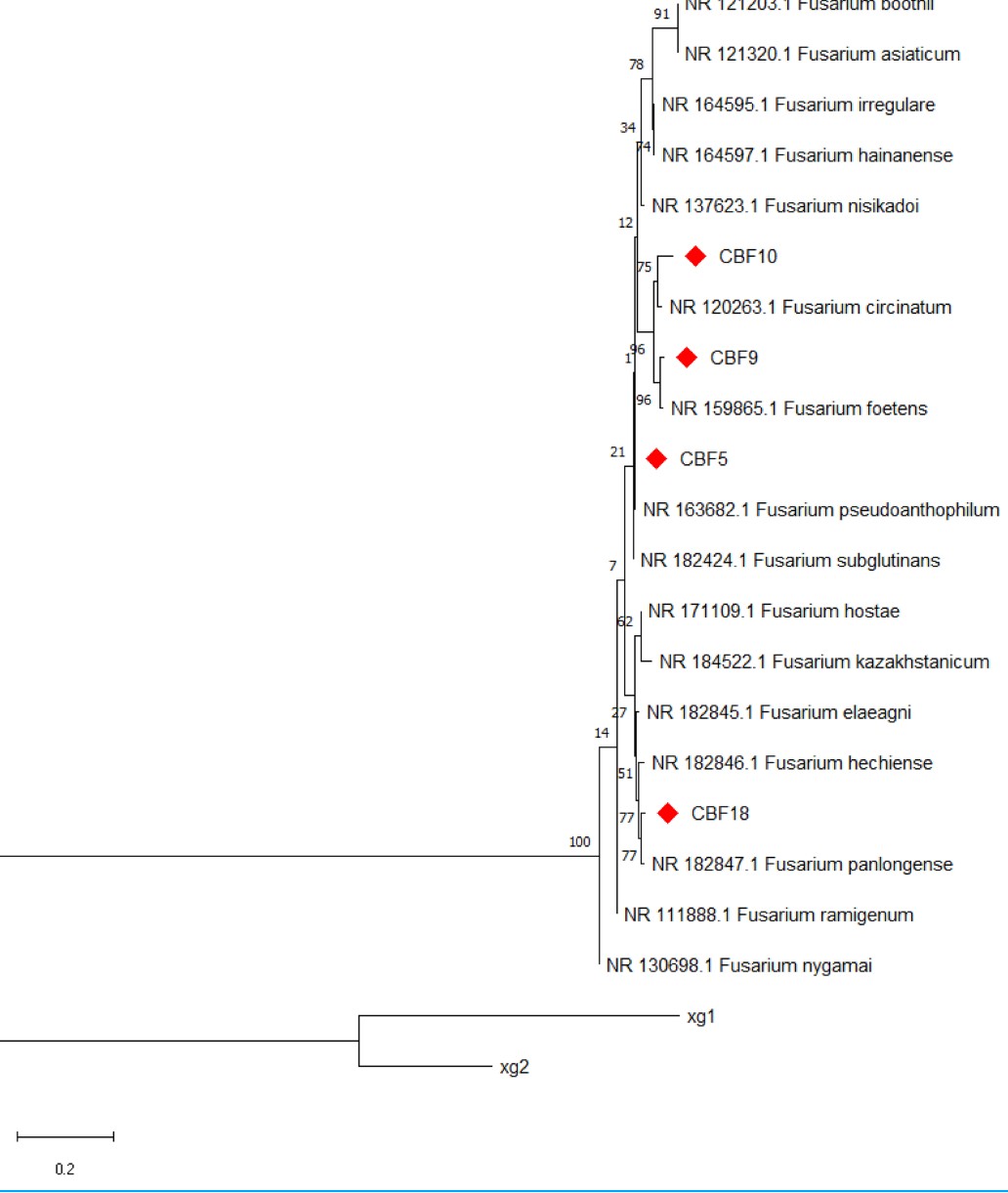

**Figure 2** Neighbor-joining tree based on ITS sequences of phenols-producing endophytic fungi. Numbers at nodes indicate bootstrap support values from 1,000 replicates. Strains xg1 and xg2 were designated as the outgroup.

extract of *F. circinatum* reached 3.0 mg/mL, there were no significant differences compared with Vc.

## Antibacterial activity

The antibacterial properties of extracts from *F. pseudoanthophilum*, *F. foetens*, *F. circinatum* and *F. panlongense* were evaluated against four typical bacteria, with specific results presented in Tables 3 and 4.

**Table 1 Determination of total phenol content of extracts of different endophytic fungi.**

| Strain | TPC | | | |
|---|---|---|---|---|
| | Ethyl acetate (mg GAE/g) | N-butanol (mg GAE/g) | Petroleum ether (mg GAE/g) | Chloroform (mg GAE/g) |
| *Fusarium pseudoanthophilum* | 60.56 ± 0.19a | 42.66 ± 0.74b | 24.87 ± 0.74c | 3.75 ± 0.19d |
| *Fusarium foetens* | 71.17 ± 0.81a | 61.31 ± 1.22b | 30.33 ± 0.67c | 13.72 ± 1.13d |
| *Fusarium circinatum* | 77.17 ± 1.93a | 66.35 ± 1.34b | 33.01 ± 0.10c | 14.36 ± 1.34d |
| *Fusarium panlongense* | 60.45 ± 0.85a | 30.87 ± 0.56b | 19.19 ± 0.98c | 9.00 ± 0.85d |

Note:
Data were analyzed using one-way ANOVA followed by *post-hoc* comparisons. Subsequent data were analyzed using the same statistical framework. a–d indicate significant differences ($p < 0.05$) between solvent treatments within the same microbial strain.

*F. pseudoanthophilum*, *F. foetens*, *F. circinatum* and *F. circinatum* showed antibacterial effects against tested four bacteria (Table 3). At the same time, The MIC of different extracts ranged from 0.5–2 mg/mL, demonstrating antibacterial activity. For *F. pseudoanthophilum*, just the ethyl acetate extracts showed antibacterial effect against four bacteria with MIC of 0.5, 2, 0.5, and 2 mg/mL, respectively. For *F. foetens* and *F. circinatum*, both ethyl acetate and n-butanol extracts showed antibacterial effect against the tested bacteria. For *F. panlongense*, except for the petroleum ether extract, all other extracts showed antibacterial effect against tested bacteria. The results indicated that the compounds extracted with ethyl acetate exhibit stronger antibacterial activity.

The MBC of the fungal extracts are summarized in Table 4. Four strains' ethyl acetate extracts demonstrated bactericidal action against *P. aeruginosa*, *E. coli* and *S. aureus*, with MBC ranging from 1.0–2.0 mg/mL.

## Compounds of endophytic fungal extracts

Four strains' ethyl acetate extracts showed potent antimicrobial and antioxidant properties, together with significant phenols content, leading to their selection for chemical analysis using LC-MS. The relevant information on the identified chemical compounds in the fungal extracts is shown in Table 5. The identified compounds included phenols (2-hydroxybenzyl alcohol, tyrosol); phenolic acids (caffeic acid, gentisic acid, *etc.*); organic acids (salicylic acid, taurine, L-lactic acid, *etc.*); fatty acids (linoleic acid, oleic acid, palmitic acid, *etc.*); short peptides. The chromatograms are shown in Fig. S3. As shown in Table 5, 28 compounds were identified from *F. pseudoanthophilum*, the main component of which was gentisic acid 570.61751 ng/mL. 20 compounds were identified from *F. foetens*, the main component of which was tyrosol (626.18840 ng/mL). This was followed by *F. circinatum* extract, which identified 23 compounds, the main component of which was tyrosol (379.54340 ng/mL). A total of 24 compounds were identified in *F. panlongense*, the main component of which was indole-3-acetic acid (476.1882 ng/mL). Overall, phenolic acids, phenols, and organic acids are the predominant components in the four endophytic fungal extracts, perhaps linked to their biological activity.

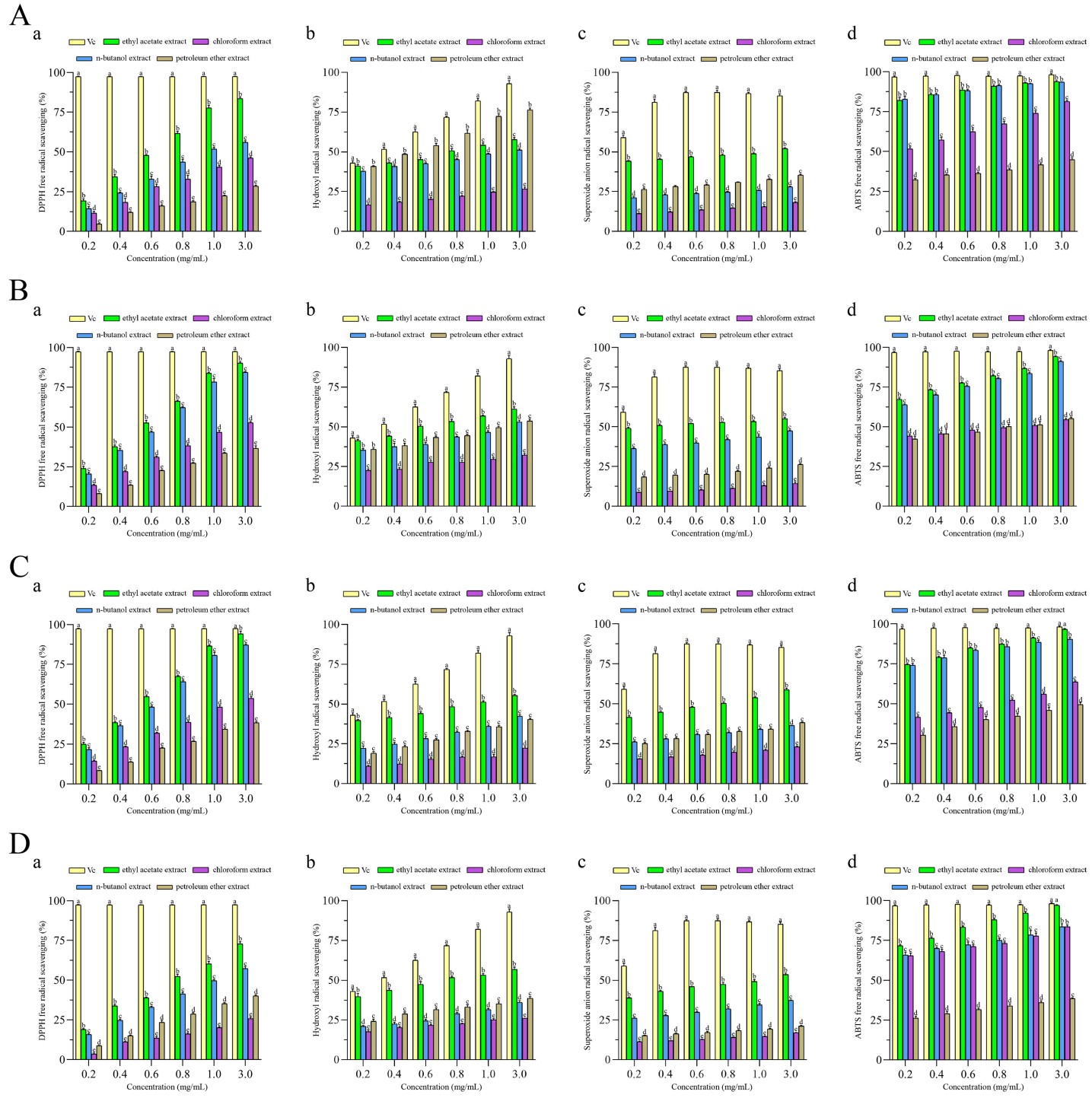

**Figure 3** **Antioxidant activities of extracts from (A)** *Fusarium pseudoanthophilum,* **(B)** *F. foetens,* **(C)** *F. circinatum,* **and (D)** *F. panlongense.* a: DPPH radical scavenging activity; b: Hydroxyl radical scavenging activity; c: Superoxide anion radical scavenging activity; d: ABTS radical scavenging activity. Letters a–e indicate significant differences among groups ($p < 0.05$).

**Table 2 Assessment of antioxidant activity of phenol-producing endophytic fungal extracts.**

| Extracts | IC$_{50}$ (mg/mL) | | | |
|---|---|---|---|---|
| | DPPH radical | Hydroxyl radical | Superoxide anion radical | ABTS radical |
| *F. pseudoanthophilum* | | | | |
| Vc | – | 0.312 ± 0.013c | 0.112 ± 0.012b | – |
| Ethyl acetate | 0.578 ± 0.150c | 0.854 ± 0.088b | 1.596 ± 0.211a | 0.012 ± 0.004b |
| n-Butanol | 1.482 ± 0.680b | 2.018 ± 0.217a | nd | 0.009 ± 0.005c |
| Chloroform | 2.915 ± 0.150a | nd | nd | 0.203 ± 0.022a |
| Petroleum ether | nd | 0.380 ± 0.012c | nd | nd |
| *F. foetens* | | | | |
| Vc | – | 0.312 ± 0.013c | 0.112 ± 0.012b | – |
| Ethyl acetate | 0.500 ± 0.012c | 0.593 ± 0.020c | 0.266 ± 0.013a | 0.078 ± 0.006b |
| n-Butanol | 0.567 ± 0.002b | 2.003 ± 0.158a | nd | 0.090 ± 0.002b |
| Chloroform | 1.896 ± 0.055a | nd | nd | 0.959 ± 0.090a |
| Petroleum ether | nd | 1.582 ± 0.128b | nd | 0.927 ± 0.201a |
| *F. circinatum* | | | | |
| Vc | – | 0.312 ± 0.013b | 0.112 ± 0.012b | – |
| Ethyl acetate | 0.481 ± 0.008b | 1.171 ± 0.054a | 0.748 ± 0.035a | 0.050 ± 0.001c |
| n-Butanol | 0.541 ± 0.003b | nd | nd | 0.026 ± 0.002d |
| Chloroform | 1.796 ± 0.081a | nd | nd | 0.625 ± 0.039b |
| Petroleum ether | nd | nd | nd | 2.468 ± 0.454a |
| *F. panlongense* | | | | |
| Vc | – | 0.312 ± 0.013b | 0.112 ± 0.012b | – |
| Ethyl acetate | 0.479 ± 0.013b | 0.840 ± 0.076a | 1.409 ± 0.189a | 0.073 ± 0.003a |
| n-Butanol | 1.931 ± 0.068a | nd | nd | 0.037 ± 0.002b |
| Chloroform | nd | nd | nd | 0.044 ± 0.013b |
| Petroleum ether | nd | nd | nd | nd |

**Note:**
a–d indicate significant differences ($p < 0.05$) between solvent treatments within the same microbial strain and antioxidant test. nd, not detected (the result higher 4 mg/mL).

# DISCUSSION

*C. blinii* is a traditional medicinal plant that contains a large number of natural active metabolites. Endophytic fungi with long-term symbiosis with host plants can produce bioactive metabolites similar to host plants (*Alam et al., 2021*). In this study, 20 endophytic fungi were isolated from different tissues of *C. blinii* to expand the natural sources of bioactive metabolites.

Phenolic compounds, which are among the primary secondary metabolites found in plants, represent a diverse group of compounds characterized by the presence of aromatic rings containing hydroxyl or methoxyl groups. They mainly include anthocyanins, tannins, flavonoids and phenolic acids. Natural phenolic compounds are of interest due to their many positive biological features, such as antioxidants, antimicrobial and anti-inflammatory activities (*Mandal, Dias & Franco, 2017*; *Jakobek & Blesso, 2023*). In our

**Table 3 Minimum inhibitory concentration (MIC) (mg/mL) of phenol-producing endophytic fungal extracts.**

| Extracts | Gram-positive bacteria | | Gram-negative bacteria | |
|---|---|---|---|---|
| | *S. aureus* | *B. subtilis* | *E. coli* | *P. aeruginosa* |
| *F. pseudoanthophilum* | | | | |
| Ethyl acetate | 0.5 | 2 | 0.5 | 2 |
| n-Butanol | 1 | nd | 0.5 | nd |
| Chloroform | 2 | nd | 2 | nd |
| Petroleum ether | 2 | nd | nd | nd |
| *F. foetens* | | | | |
| Ethyl acetate | 0.5 | 2 | 0.5 | 1 |
| n-Butanol | 1 | 2 | 0.5 | 2 |
| Chloroform | 2 | 2 | 2 | nd |
| Petroleum ether | nd | nd | 2 | nd |
| *F. circinatum* | | | | |
| Ethyl acetate | 1 | 1 | 2 | 1 |
| n-Butanol | 2 | 2 | 1 | 2 |
| Chloroform | nd | nd | 1 | nd |
| Petroleum ether | nd | 2 | 2 | nd |
| *F. panlongense* | | | | |
| Ethyl acetate | 1 | 2 | 0.5 | 1 |
| n-Butanol | 1 | 1 | 1 | 0.5 |
| Chloroform | nd | 2 | 2 | 2 |
| Petroleum ether | nd | nd | nd | nd |

**Note:**
nd, not detected (result higher 3.00 mg/mL).

research, four strains of endophytic fungi producing phenolic compounds were screened, all of which belonged to *Fusarium*. *Caicedo et al. (2019)* isolated *F. oxysporum* from *Otoba gracilipes* and found that the extracts scavenged 51.5% of DPPH within 5 min of the reaction. Therefore, the four endophytic fungi we isolated have good research and application value.

Among the four endophytic fungi strains, *F. circinatum* had the highest TPC, and its antioxidant capacity was also the strongest among the four strains, followed by *F. foetens*. In addition, the antioxidant activity of other strains also seemed to be positively correlated with the total phenols content. *Moreno Gracia et al. (2021)* found that the total phenolic compound content in various almond varieties was strongly correlated with their total antioxidant activity. Moreover, *César et al. (2022)* obtained 12 chromatographic fractions from the methanol extract of *Litsea glaucescens* through a series of polar gradient elution. Among these fractions, those exhibiting the highest antioxidant activity also demonstrated the highest TPC. All these indicate that phenolic compounds have good antioxidant properties. At a concentration of 3.0 mg/mL, the endophytic fungus extracts showed better scavenging ability against DPPH and ABTS radicals than hydroxyl and superoxide anion radicals in our investigation. In addition, the ABTS radical clearance rate of *F. circinatum*,

**Table 4 Minimum bactericidal concentration (MBC) (mg/mL) of phenol-producing endophytic fungal extracts.**

| Extracts | Gram-positive bacteria | | Gram-negative bacteria | |
|---|---|---|---|---|
| | *S. aureus* | *B. subtilis* | *E. coli* | *P. aeruginosa* |
| *F. pseudoanthophilum* | | | | |
| Ethyl acetate | 2 | nd | 2 | 2 |
| n -Butanol | nd | nd | 2 | nd |
| Chloroform | nd | nd | nd | nd |
| Petroleum ether | nd | nd | nd | nd |
| *F. foetens* | | | | |
| Ethyl acetate | 2 | nd | 2 | 1 |
| n-Butanol | 2 | nd | 1 | 2 |
| Chloroform | nd | nd | nd | nd |
| Petroleum ether | nd | nd | nd | nd |
| *F. circinatum* | | | | |
| Ethyl acetate | 2 | nd | 2 | 1 |
| n-Butanol | nd | nd | 1 | nd |
| Chloroform | nd | nd | nd | nd |
| Petroleum ether | nd | nd | nd | nd |
| *F. panlongense* | | | | |
| Ethyl acetate | 2 | nd | 2 | 1 |
| n-Butanol | nd | nd | nd | nd |
| Chloroform | nd | nd | nd | nd |
| Petroleum ether | nd | nd | nd | nd |

**Note:**
nd, not detected (result higher 3.00 mg/mL).

which exhibited the highest antioxidant activity, and that of Vc showed no significant difference ($p < 0.05$). However, the antioxidant activity of these four strains of endophytic fungi was lower than Vc, probably because the extracts of endophytic fungi are a complex of multiple bioactive substances. However, *Kumari et al. (2021)* purified the extract of *Penicillium citrinum* isolated from *Azadirachta indica* and found that the pure fraction exhibited no antioxidant activity, whereas the crude extract demonstrated significant antioxidant activity. Therefore, in this research, four endophytic fungi ethyl acetate extracts were analyzed by LC-MS to find the main metabolites that exert antioxidant effects in them. This will pave the way for whether to purify metabolites in the future.

At the same time, endophytic fungi that have long lived in symbiosis with plants are also able to secrete metabolites with antimicrobial activity. For example, *Wang et al. (2023b)* isolated 54 strains from Gannan navel orange, which had excellent antibacterial activity against methicillin-resistant *S. aureus*, *E. coli* and *Xanthomonas citri* subsp. In our research, four endophytic fungi exhibited antibacterial activity against four pathogens. Among them, extracts of *F. foetens* had the best antibacterial and bactericidal effect, followed by *F. circinatum*. The antibacterial mechanism of phenolic compounds mainly includes the inhibition of the synthesis of biomacromolecules, the destruction of bacterial

**Table 5 The identification of the chemical composition of endophytic fungal extracts by LC-MS analysis.**

| S/N | Name of identified compound | Adducts | RT (min) | Molecular formula | M/S | Endophytic fungal extracts (ng/mL) | | | |
|---|---|---|---|---|---|---|---|---|---|
| | | | | | | F. pseudoanthophilum | F. foetens | F. circinatum | F. panlongense |
| 1 | Caffeic acid | [M-H]⁻ | 4.1 | $C_9H_8O_4$ | 179.0345 | 84.20629 | 83.40548 | 90.98647 | 66.53597 |
| 2 | Gentisic acid | [M-H]⁻ | 4.7 | $C_7H_6O_4$ | 153.0186 | 570.61751 | 144.21708 | 103.15063 | 128.77121 |
| 3 | Homovanillic acid | [M-H]⁻ | 7.0 | $C_9H_{10}O_4$ | 181.0502 | 218.52664 | nd | nd | nd |
| 4 | (Z)-9,12,13-trihydroxyoctadec-15-enoic acid | [M-H]⁻ | 7.6 | $C_{18}H_{34}O_5$ | 329.234 | nd | nd | 47.00341 | 25.58543 |
| 5 | 12-Hydroxyoctadecanoic acid | [M-H]⁻ | 11.7 | $C_{18}H_{36}O_3$ | 299.2596 | 278.40240 | 360.42660 | 232.45480 | 138.16360 |
| 6 | 16-Hydroxyhexadecanoic acid | [M-H]⁻ | 10.4 | $C_{16}H_{32}O_3$ | 271.2284 | 42.40630 | 107.53030 | 102.33620 | 36.90116 |
| 7 | 2,6-Dihydroxy-4-Methoxytoluene | [M-H]⁻ | 4.8 | $C_8H_{10}O_3$ | 153.0549 | nd | 26.89187 | 144.21370 | nd |
| 8 | 2-Hydroxybenzyl alcohol | [M-H]⁻ | 3.5 | $C_7H_8O_2$ | 123.0442 | 39.06816 | nd | nd | nd |
| 9 | 2-Methylglutaric acid | [M-H]⁻ | 2.0 | $C_6H_{10}O_4$ | 145.0499 | 46.80742 | nd | nd | nd |
| 10 | 3-Hydroxyphenylacetic acid | [M-H]⁻ | 4.7 | $C_8H_8O_3$ | 151.0393 | 22.90005 | nd | nd | nd |
| 11 | 3-Hydroxypicolinic acid | [M-H]⁻ | 1.8 | $C_6H_5NO_3$ | 138.0189 | 29.10585 | 26.07315 | 27.56419 | 28.38109 |
| 12 | 3-Methoxyphenylacetic acid | [M-H]⁻ | 5.5 | $C_9H_{10}O_3$ | 165.0551 | nd | nd | nd | 38.88268 |
| 13 | 3-Phenyllactic acid | [M-H]⁻ | 4.9 | $C_9H_{10}O_3$ | 165.0551 | 74.91138 | nd | nd | nd |
| 14 | 6-Hydroxycaproic acid | [M-H]⁻ | 4.4 | $C_6H_{12}O_3$ | 131.0705 | 31.99286 | nd | nd | nd |
| 15 | 9-HODE | [M-H]⁻ | 10.6 | $C_{18}H_{32}O_3$ | 295.2284 | 54.83800 | nd | nd | 49.78712 |
| 16 | 9-HpODE | [M-H]⁻ | 8.5 | $C_{18}H_{32}O_4$ | 311.2234 | 31.02287 | 32.76858 | 39.72435 | 28.06550 |
| 17 | Ala-Phe | [M-H]⁻ | 7.2 | $C_{12}H_{16}N_2O_3$ | 235.1089 | 160.47810 | nd | nd | 91.16756 |
| 18 | Benzoic acid | [M-H]⁻ | 4.6 | $C_7H_6O_2$ | 121.0285 | 93.83307 | 30.30524 | 37.99612 | 70.63667 |
| 19 | FA 13:3+1O | [M-H]⁻ | 8.4 | $C_{13}H_{20}O_3$ | 223.134 | nd | nd | nd | 105.80480 |
| 20 | FA 18:1+3O | [M-H]⁻ | 7.1 | $C_{18}H_{34}O_5$ | 329.234 | 30.38364 | nd | 49.65539 | 28.78666 |
| 21 | FA 9:1+1O | [M-H]⁻ | 6.1 | $C_9H_{16}O_3$ | 171.1022 | 40.23517 | 33.76572 | 135.51830 | nd |
| 22 | Glycerol 3-phosphate | [M-H]⁻ | 0.8 | $C_3H_9O_6P$ | 171.0059 | 48.92283 | 31.73365 | 32.11357 | 32.42081 |
| 23 | Hydroxyphenyllactic acid | [M-H]⁻ | 5.8 | $C_9H_{10}O_4$ | 181.0502 | nd | 46.55864 | 65.65048 | nd |
| 24 | Imidazoleacetic acid | [M-H]⁻ | 1.7 | $C_5H_6N_2O_2$ | 125.0347 | 61.34382 | 60.65119 | 72.15062 | 34.30137 |
| 25 | Indole-3-acetic acid | [M-H]⁻ | 5.8 | $C_{10}H_9NO_2$ | 174.0555 | nd | nd | 72.09359 | 476.18820 |
| 26 | Linoleic acid | [M-H]⁻ | 13.1 | $C_{18}H_{32}O_2$ | 279.2333 | 49.63264 | 44.75582 | 41.16748 | 53.13914 |
| 27 | L-lactic acid | [M-H]⁻ | 0.9 | $C_3H_6O_3$ | 89.02324 | 102.27420 | 82.83572 | 186.78670 | 63.88027 |
| 28 | LysoPA (i-12:0/0:0) | [M-H]⁻ | 9.6 | $C_{15}H_{31}O_7P$ | 353.1742 | 121.36250 | 128.17770 | 142.56070 | 142.88170 |
| 29 | LysoPE (14:0/0:0) | [M-H]⁻ | 8.0 | $C_{19}H_{40}NO_7P$ | 424.248 | 113.59740 | 116.88210 | 126.65270 | 118.61340 |
| 30 | Pyruvic acid | [M-H]⁻ | 1.3 | $C_3H_4O_3$ | 87.00764 | nd | nd | 75.49413 | 125.22440 |
| 31 | Salicylic acid | [M-H]⁻ | 3.9 | $C_7H_6O_3$ | 137.0236 | 115.78080 | 31.46453 | 42.58878 | 34.82906 |
| 32 | Taurine | [M-H]⁻ | 0.8 | $C_2H_7NO_3S$ | 124.0064 | 95.86946 | nd | nd | nd |
| 33 | Tyrosol | [M-H]⁻ | 6.1 | $C_8H_{10}O_2$ | 137.0599 | 38.28279 | 626.18840 | 379.54340 | nd |
| 34 | Oleic acid | [M-H]⁻ | 13.9 | $C_{18}H_{34}O_2$ | 281.2489 | 47.52210 | 25.44637 | nd | 37.38932 |
| 35 | Palmitic acid | [M-H]⁻ | 13.7 | $C_{16}H_{32}O_2$ | 255.2332 | 33.79977 | 32.26168 | 34.02589 | 38.75807 |

**Note:**
nd, not detected (result lower 20 ng/mL).

cellular structures, and the suppression of key metabolic pathways (*Tang et al., 2024*). The four endophytic fungi ethyl acetate extracts were mainly composed of phenolic compounds, including homovanilic acid, tyrosol and caffeic acid. Therefore, it may be that phenolic compounds give them antibacterial properties. Current research further substantiates the potential of *Fusarium* species as prolific producers of antimicrobial metabolites. For instance, *Ariantari et al. (2021)* conducted antimicrobial assays on lateropyrone, a metabolite isolated from the endophytic fungus *Fusarium* sp. BZCB-CA, and found that it exhibited inhibitory activity against *S. aureus* with a MIC of 3.1 µmol/L. Similarly, *Khattak et al. (2024)* purified extracts from the endophytic fungus *F. oxysporum*, isolated from the medicinal plant *Myrtus communis*, and observed that the purified extracts inhibited *P. aeruginosa* at varying concentrations, with inhibition rates ranging from 14% to 41%. Our findings reinforce the antimicrobial potential of metabolites derived from *Fusarium* species.

The compounds in the extracts of the endophytic fungi were identified using LC-MS. In addition to phenolic compounds, we found that *F. pseudoanthophilum*, *F. foetens*, *F. circinatum* and *F. panlongense* could produce other bioactive metabolites. All four endophytic fungi produced a large amount of LysoPA and LysoPE. Both LysoPA and LysoPE belong to a molecular family called glycero-lysophospholipids (*Grzelczyk & Gendaszewska-Darmach, 2013*; *Makide et al., 2009*). LysoPA has been most studied in oncology, and has been reported to positively impact the genesis and progression of large of tumors (*Aiello & Casiraghi, 2021*; *Ray et al., 2020*). Otherwise, LysoPE not only prolongs the freshness of fruit (*Jung et al., 2019*), but also enhances the basal resistance of plants against certain pathogens (*Völz et al., 2021*). Moreover, *F. panlongense* can generate large amounts of indole-3-acetic acid (IAA). IAA plays a variety of roles in plant development, including branch growth, root development and fruit maturation (*Luo, Zhou & Zhang, 2018*). All these indicate that endophytic fungi have potential applications in health care, medicine and agriculture. Moreover, because endophytic fungi have the advantages of simple fermentation, fast growth and less pollution (*Ludwig-Müller, 2015*), they can reduce the damage to endangered plants and environmental pollution caused by chemical synthesis (*Liu et al., 2021*). Therefore, endophytes isolated from *C. blinii* have great potential for research and application.

## CONCLUSIONS

In conclusion, phenols-producing endophytic fungi were screened from *C. blini*. In addition, the different extracts of endophytic fungi were confirmed to have good antioxidant and antibacterial activities. At the same time, LC-MS analysis found that their antioxidant activity and antibacterial activity were mainly related to the phenolic compounds. In the end, this research shows that four phenols-producing endophytic fungi from *C. blinii* have great potential for research and application. However, in order to realize the application of these fungi, it is necessary to isolate and purify the main compound.

### Funding

This study was supported by the Technology Department of Sichuan Province International Cooperation Program of China (Grant numbers 2023YFH0043); Sichuan Sharing and Service Platform of Scientific and Technological Resource (Enzyme Resource) of China (Grant numbers 2020JDPT0018); the National Key R&D Program of China (Grant numbers 2021YFD1200105); Sichuan Science and Technology Program (Grant numbers 2024ZDZX0057) and the International Science and Technology Innovation Cooperation/Hong Kong, Macao, and Taiwan Science and Technology Innovation Cooperation Project of Sichuan Province (2025YFHZ0143). The funders had no role in study design, data collection and analysis, decision to publish, or preparation of the manuscript.

### Grant Disclosures

The following grant information was disclosed by the authors:
Technology Department of Sichuan Province International Cooperation Program of China: 2023YFH0043.
Sichuan Sharing and Service Platform of Scientific and Technological Resource (Enzyme Resource) of China: 2020JDPT0018.
National Key R&D Program of China: 2021YFD1200105.
Sichuan Science and Technology Program: 2024ZDZX0057.
International Science and Technology Innovation Cooperation/Hong Kong, Macao, and Taiwan Science and Technology Innovation Cooperation Project of Sichuan Province: 2025YFHZ0143.

### Competing Interests

The authors declare that they have no competing interests.

### Author Contributions

- Yujie Jia conceived and designed the experiments, performed the experiments, analyzed the data, prepared figures and/or tables, and approved the final draft.
- Guodong Zhang conceived and designed the experiments, authored or reviewed drafts of the article, and approved the final draft.
- Qiqi Xie performed the experiments, analyzed the data, prepared figures and/or tables, and approved the final draft.
- Jiwen Tao performed the experiments, prepared figures and/or tables, and approved the final draft.
- Tongliang Bu analyzed the data, authored or reviewed drafts of the article, and approved the final draft.
- Xinyu Zhang performed the experiments, prepared figures and/or tables, and approved the final draft.

- Yirong Xiao conceived and designed the experiments, authored or reviewed drafts of the article, and approved the final draft.
- Zhao Chen conceived and designed the experiments, authored or reviewed drafts of the article, and approved the final draft.
- Qingfeng Li analyzed the data, authored or reviewed drafts of the article, and approved the final draft.
- Zizhong Tang conceived and designed the experiments, authored or reviewed drafts of the article, and approved the final draft.

## DNA Deposition

The following information was supplied regarding the deposition of DNA sequences:

The sequences are available at GenBank: PQ780195 to PQ780198.

## Data Availability

The raw data and sequences are available in the Supplemental Files.

## Supplemental Information

Supplemental information for this article can be found online at http://dx.doi.org/10.7717/peerj.19464#supplemental-information.

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
