# Peer review of "Isolation and identification of endophytic fungi from Conyza blinii that exhibit antioxidant and antibacterial activities"

_PeerJ, doi:10.7717/peerj.19464_

## Round 0.1 · original submission · Major Revisions

We have now received and evaluated the comments from two reviewers. While both reviewers acknowledge the potential value of your study, they have raised significant concerns that must be addressed before further consideration for publication. Based on their assessments, we are requesting a major revision of your manuscript. Below, we summarize the key issues identified, along with specific points that require your attention:
1. Language and Technical Presentation
• The manuscript requires substantial improvement in English to ensure clarity and readability. We strongly recommend seeking assistance from a proficient English speaker or a professional editing service.
• Some figure and table legends lack sufficient descriptions; they should clearly explain the presented data.
• Formatting inconsistencies, including incorrect spacing and typographical errors, need to be corrected throughout the manuscript
2. Justification and Study Design
• The rationale for selecting C. blinii endophytic fungi needs clarification and improve previous studies and provide appropriate references.
• The manuscript should include a more detailed description of the C. blinii plants used in the study: number, phenological stage, ecological conditions, observed disease symptoms or insect presence?.
• There is ambiguity regarding the number of plants sampled. Please clarify.
3. Methodological Concerns
There are several methodological concerns raised by both reviewers. Please clarify these points:
• Fungal Isolation and Selection
• Identification of Fungal Isolates
• Normalization of Cultures and Optimization of Culture Conditions
4. Data Presentation and Accuracy
There are several comments to improve data presentation on Tables and Figures raised by both reviewers. Please address them.
5. Discussion and Interpretation of Findings
• The Discussion should incorporate a broader comparison of results with previously published studies on similar fungal species.
• The metabolic profiles and bioactive compound production of the isolates should be discussed in the context of relevant literature.
• The biological significance of the findings should be explored in more depth.
• The conclusions should provide a clearer perspective on potential applications of these fungi and their metabolites, addressing both benefits and potential limitations (e.g., mycotoxin production by Fusarium species).

Reviewer 1 ·

Basic reporting

Some parts of this article fail to meet technical standards. The article does not share some relevant data and fails to correctly describe hypothesis, interpret the results and compare them using the published literature.The figure and table legends do not provide suitable descriptions of the data presented. Please, see additional comments.

Experimental design

Some parts of this investigation were not performed following a high technical standard. Methods are not described with sufficient detail. Please, see additional comments.

Validity of the findings

In this study 20 endophytic fungal isolates were purified from different tissues from an unknown number of Conyza blinii plants. The number of different fungal species isolated is unknown. The ecological characteristics of the Conyza collection site is unknown. A preliminary visual screening of color intensity obtained from a reaction between the fermentation broth filtered from a liquid culture of each of the twenty uncharacterized isolates and a chromogenic reagent was performed. It seems that the quantification of fungal density on the liquid culture used to obtain the fermentation broth was not performed. Lack of normalization among cultures does not allow comparisons among isolates for their phenol producing capabilities. Based on this preliminary visual screening of color intensity, four isolates were selected for species identification and further studies on their phenolic production capability, antioxidant and antibacterial activities. The four selected strains were identified as Fusarium species using ITS sequencing but the species identification obtained was not confirmed with sequencing of other genes usually preferred for the precise identification of Fusarium species (such as a translation elongation factor gene). Sixteen of the twenty fungal isolates obtained in this work were not identified and proper optimization of growing conditions (temperature, nutritive media, number of days of cultivation) conducive to maximum phenolic production for each fungal species was not performed. The approach adopted in this work might have overlooked unknown fungal species with interesting capabilities in addition to the four selected Fusarium isolates. In my opinion, given the small scale of this work, species identification should have been performed first for the twenty fungal isolates obtained. Then, growth conditions should have been optimized for each species, and fungal density normalized across fungal cultures to select the best isolates.

Additional comments

Line 33, 36 : Check the journal's author guidelines for the use of abbreviations and acronyms in the abstract without defining them the first time of use.
Line 42: Species names should be italicized.
Line 45: Discussion section of the abstract is insufficient. This short paragraph seems more like a conclusion than a discussion.
Line 48: Keywords. Avoid the duplication of words already in the title.
Lines 53, 54, 56…: Add space before the parentheses. Check the entire manuscript.
Line 81. Effective
Line 81. Antimicrobial bioactivity.
Line 92-93. “At present, the researches on endophytic fungi of C. blinii are few and single.”. Please rephrase.
Line 100-102: Conyza material should be described (number of Conyza plants collected, phenological stage, environment in which the plants were collected, presence of disease symptoms, presence of insects or other pests etc.)
Line 105: “The plant was divided.”.: Was only one Conyza plant used in this study? According to line 100 “ C. blinii were taken” several plants were taken.
Line 111-112: correct: (add 50 μg/L kanamycin sulfate and ampicillin, the following is the same),
Line 117. Please specify PDB volume used. Please specify if fungal density in the liquid culture was quantified to allow comparability between phenol production capacity among fungal isolates.
Line 125. Correct parenthesis.
Line 134. “A large number of phenols-producing endophytic fungi” What does large number mean? Given the fact that only 4 isolates were used, does it refer to a large volume of liquid culture for each of the four fungal isolates. Please specify the volume of fungal culture used.
Line 135-137. “The fermentation broth underwent two extractions with ethyl n-butanol, acetate, choroform, and petroleum” Does it mean two extractions with each solvent? Was each solvent used twice in sequential order on the same aliquot of fermentation broth to obtain molecules with different polarity?. Was each solvent used twice on independent fermentation broth aliquots to compare the extraction efficacy of each solvent?.
Line 136. Please correct: “ethyl n-butanol, acetate” by " n-butanol, ethyl acetate".
Line 137. “with each solvent receiving a volume equal to twice that of the fermentation broth.” Please, specify the volume of fermentation broth used for each extraction.
Line 135-140. “the crude extracts were subjected to freeze-drying using lyophilization”. Was lyophilized the organic phase obtained with each solvent? Extracts obtained with the organic solvents are usually dried with rotavapor or similar method.
Line 141, 142: correct “Phenolsic “
Line 144. Introduce the acronym FC in line 142
Line 152: Introduce the acronyms DPPH and ABTS
Line 154. Please replace “Vc (vitamin C)” by vitamin C (VC). Ascorbic acid?. Suppliers of chemicals used in this work could be mentioned.
Line 204. Correct “MIC Among “
Line 230. If the twenty fungal isolates were not identified through the use of ITS sequencing, nor observation of microscopic morphology, how do the authors know that they were distinct?
Line 231. “8 isolates were obtained from roots, 7 from stems, and 5 from leaves” Please, specify number of different Conyza plants for each Conyza organ.
Line 232. “The endophytic fungi were extracted from different tissues of C. blinii are listed in Supplementary Table 4” Only codes are listed in Supplementary Table 4. Species were not identified. In Supplementary Table 4: please correct “Endopyte”. The column “total” does not provide relevant information, I suggest to replace it for the number of conyza plants from which the fungal isolates were obtained. (see comment above regarding line 231)
Line 238. “all strains had the ability to generate phenols”. Sixteen of the twenty fungal isolates obtained in this work were not identified and optimization of their growing conditions (temperature, nutritive media, days of cultivation) conducive to maximum phenolic production, that can vary among fungal species, was not performed. In addition, without quantification of fungal density for normalization across cultures, the comparability of phenol producing capabilities made among isolates may not be correct.
Line 241-248. The preliminary species identification of the four Fusarium isolates made in this work with ITS sequencing should be confirmed using additional primers such as those specific for a translation elongation factor gene, usually preferred for Fusarium species.
Line 250-255. I suggest to add an additional supplementary figure with the analysis of each extract on TLC plates. The information provided by this suggested TLC figure could be more relevant that the information provided for other figures presented such as supplementary figure 2 (agarose gel electrophoresis of fungal ITS sequences).
Line 251 and 253. Please correct “total phenolic content (TPC)”
Line 252. Please, describe in the text the meaning of “mg GAE/g”: milligrams of gallic acid equivalents per gram of extract.
Line 252-253. Please, specify the yield of the extraction (%), the volume of fermentation broth, the extract weight...
Line 253-255. Please rephrase.
Line 306-308. The high tyrosol producers CBF9 (Fusarium foetens) and CBF10 (Fusarium circinatum) were both collected from the stem of Conyza. Could this have a biological significance?
Table 1. (line 2). “a-d: significant differences between different groups (p < 0.05).”: between different solvents? Please add the statistical test used.

Reviewer 2 ·

Basic reporting

The English language should be improved to ensure that an international audience can clearly understand your text. I suggest you have a colleague who is proficient in English and familiar with the subject matter review your manuscript, or contact a professional
editing service.
In the Introduction the authors stated "This study might potentially facilitate the future use of bioactive compounds generated by C. blinii endophytic fungi in the food or medicinal sectors". Detailed explanations for these perspectives should be provided.
The Discussion section needs more details. Relevant and recent studies on the isolated and selceted fungi should be cited and discussed. Particular attention must be given to the metabolites produced by these fungi.
The quality of the figures must be increased.
Different decimals are reported in Table 5 the authors should report the same number of decimals.

Experimental design

Why the auhtors decided to work on endophytic fungi of C. blinii? They stated "At present, the researches on endophytic fungi of C. blinii are few and single." What does it means? Is there a single research or few? The relative references are missing.

Validity of the findings

From table six in the supporting information it can be seen that the masses were also recorded in positive mode but only the results of negative mode were reported. What about all the other compounds produced by the fungi? Fusarium spp. are able to produce micotoxins and it is important to understand if the selected fungi are also producers of micotoxins. This will help to propose them as producers of compounds with potential medicinal value.
The retention times reported in Table 5 are not coincident with those reported in Supplementary Figure 3.
Were internal standards used for the quantification of the metabolites present in the different extracts?
Are the authors certain that all the metabolites were produced by fungi or are some of them the result of contamination? For example 4-dodecylbenzenesulfonic acid is a synthetic strong anionic surfactant and it's a key ingredient in many household and industrial detergents. Are the authors sure that in not a contaminat?
The conclusions need to be implemented by adding future perspectives on the practical application of these fungi, of thier extracts and of their pure metabolites.

Additional comments

Line 2: "higher"? I suggest to delete it.
Line 22-23: Why it necesseray to report this sentence "As a medicinal plant, Conyza blinii may contain a wealth of bioactive constituents"?
Line 49: "Conyza blini" should be "Conyza blinii".
Line 136: ethyl n-butanol???

---

## Round 0.2 · accepted · Accept

Thanks for taking into consideration all comments and modifications raised by both reviewers. Your manuscript is ready for publication

Reviewer 2 ·

Basic reporting

No comment

Experimental design

No comment

Validity of the findings

No comment

Additional comments

No comment